# Predictors of New Dementia Diagnoses in Elderly Individuals: A Retrospective Cohort Study Based on Prefecture-Wide Claims Data in Japan

**DOI:** 10.3390/ijerph18020629

**Published:** 2021-01-13

**Authors:** Yuriko Nakaoku, Yoshimitsu Takahashi, Shinjiro Tominari, Takeo Nakayama

**Affiliations:** 1Department of Health Informatics, Kyoto University School of Public Health, Kyoto 606-8501, Japan; takahashi.yoshimitsu.3m@kyoto-u.ac.jp (Y.T.); tominari-s@umin.ac.jp (S.T.); nakayama.takeo.4a@kyoto-u.ac.jp (T.N.); 2Department of Preventive Medicine and Epidemiology, National Cerebral and Cardiovascular Center, Suita 564-8565, Japan

**Keywords:** claims data, dementia, medical care system for the elderly, predictors, prescription

## Abstract

Preventing dementia in elderly individuals is an important public health challenge. While early identification and modification of predictors are crucial, predictors of dementia based on routinely collected healthcare data are not fully understood. We aimed to examine potential predictors of dementia diagnosis using routinely collected claims data. In this retrospective cohort study, claims data from fiscal years 2012 (baseline) and 2016 (follow-up), recorded in an administrative claims database of the medical care system for the elderly (75 years or older) in Niigata prefecture, Japan, were used. Data on baseline characteristics including age, sex, diagnosis, and prescriptions were collected, and the relationship between subsequent new diagnoses of dementia and potential predictors was examined using multivariable logistic regression models. A total of 226,738 people without a diagnosis of dementia at baseline were followed. Of these, 26,092 incident dementia cases were detected during the study period. After adjusting for confounding factors, cerebrovascular disease (odds ratio, 1.15; 95% confidence interval, 1.11–1.18), depression (1.38; 1.31–1.44), antipsychotic use (1.40; 1.31–1.49), and hypnotic use (1.17; 1.11–1.24) were significantly associated with subsequent diagnosis of dementia. Analyses of routinely collected claims data revealed neuropsychiatric symptoms including depression, antipsychotic use, hypnotic use, and cerebrovascular disease to be predictors of new dementia diagnoses.

## 1. Introduction

Dementia among elderly people is an important public health concern. The number of elderly people living with dementia exceeded 50 million worldwide in 2018, and is projected to triple by 2050 [1]. In Japan, the world fastest aging society, those aged ≥65 years and ≥75 years accounted for 27% and 13% of the population, respectively, in 2015 [2]. The Hisayama study, which comprised volunteer Japanese residents, reported that the prevalence of dementia among individuals aged ≥65 years increased from 1985 to 2012 for all-cause dementia (from 7% to 11%) and Alzheimer’s disease (from 2% to 7%) [3].

Previous studies have identified some predictors of dementia [4,5,6]. One meta-analysis has identified seven potentially modifiable predictors for dementia: diabetes mellitus, hypertension, obesity, smoking, physical inactivity, cognitive inactivity, and depression [7]. In 2020, the Lancet commission report has identified 12 modifiable risk factors for dementia: less education, hypertension, hearing impairment, smoking, obesity, depression, physical inactivity, diabetes, infrequent social contact, excessive alcohol consumption, head injury, and air pollution [6]. Several risk prediction models have been developed based on demographics, neuropsychological testing scores, and magnetic resonance imaging data [8]. However, these studies examined selected volunteers as participants and used variables that are not routinely collected in medical care. Consequently, the applicability of prediction models for real practices, as well as estimation of entire population burden, are somewhat limited [9].

Administrative claims data cover usual medical care for all insured people. Although these data are potentially useful for examining dementia predictors, previous studies on this topic have been limited. Accordingly, this study aimed to examine the usefulness of routinely obtained administrative claims data for identifying predicters of new dementia diagnoses in a Japanese elderly population.

## 2. Materials and Methods

### 2.1. Design and Data Source

This was a retrospective cohort study. In 2008, Japan initiated the medical care system for the elderly aged ≥75 years. We extracted data from an administrative claims database covering almost all residents aged ≥75 years in Niigata prefecture. Enrollment in this insurance system is mandatory for all Japanese citizens except for welfare recipients. We obtained anonymized claims data for fiscal year (FY) 2012 (baseline) and FY2016 (follow-up), which included inpatient/outpatient medical diagnoses (coded by the International Classification of Diseases 10th revision, ICD-10) and prescription drugs (categorized according to the Anatomical Therapeutic Chemical classification system, ATC code). For each person, data were merged using a unique but anonymous identification code. The follow-up period was 60 months, from April 2012 to March 2017, according to the Japanese fiscal year which starts on 1 April and ends on 31 March.

This study was approved by the ethics committee of the Kyoto University Graduate School and Faculty of Medicine (No. R1767). Participant written consent was waived because this study involved routinely collected claims data that were de-identified before provision and anonymously managed at all stages, including data cleaning and statistical analysis.

### 2.2. Study Population

Figure 1 shows a flowchart of participant selection. We selected participants from the data source according to the following criteria: (1) those residing in Niigata prefecture who were insured by the medical care system for the elderly aged ≥75 years from FY2012 to FY2016, and (2) those with no record of dementia diagnosis in FY2012. We could not obtain claims data for individuals covered by public assistance, so these individuals were not included in the study.

### 2.3. Exposure Including Predictors

We selected variables which could be obtained from the administrative claims database as potential risk factors of dementia [5,6,10]. The following predictor variables were included in the risk model:Demographic measures: age, sex.Medical diagnosis: diabetes mellitus (ICD-10: E11–E14), ischemic heart disease (I20–I25), cerebrovascular disease (I60–I69), atrial fibrillation/flutter (I48) at baseline, and current depression diagnosis (F32, F33)/treatment with antidepressants.Prescriptions: antihypertensive drugs (ATC-code: C02, C03, C07, C08, C09), antihyperlipidemia drugs (C10), antidepressants (N06A), antipsychotics (N05A), anxiolytics (N05B), hypnotics (N05C), and antithrombotics (B01AC).

Prevalent users were defined as those who had been prescribed any of the above-mentioned drugs at least once at baseline. According to Walter’s definition of predictors for dementia and availability in our database, we defined exposure variables as follows: treated with antihypertensive drugs, dyslipidemia drugs, hypnotics, antipsychotics, or anxiolytics, and received depression diagnosis/treatment with antidepressants [10].

### 2.4. Outcomes

The outcome was a new diagnosis of dementia at follow-up (FY 2016), which was determined from the following medical diagnoses coded by the ICD-10: dementia in Alzheimer’s disease (F00), vascular dementia (F01), dementia in other diseases (F02), unspecified dementia (F03), Alzheimer’s disease (G30), and other degenerative diseases of the nervous system (G31). Hessler’s criteria were used for dementia diagnoses [11]. To increase the validity of the diagnoses, dementia cases were narrowed to participants who received diagnoses in at least two billing quarters of health insurance or both in- and outpatient settings, as was done in a previous analysis of claims data [11].

### 2.5. Statistical Analysis

For the descriptive analysis, continuous variables were reported as means and standard deviations (SD), while categorical variables were reported as numbers and percentages. To examine potential predictors of a new diagnosis of all-cause dementia, we used multivariable logistic regression models. One model was adjusted for age and sex (age- and sex-adjusted model). Another model was adjusted for age, sex, and potential risk factors for dementia (medical diagnosis and/or prescriptions), including comorbidities of cerebrovascular disease, diabetes mellitus, ischemic heart disease, atrial fibrillation, and depression or use of antidepressants, and use of antipsychotics, anxiolytics, hypnotics, antihypertensive drugs, dyslipidemia drugs, and antithrombotics. In order to elucidate the aging effect of these potential predictors, we also analyzed the data by age group: 75–79 years, 80–84 years, and ≥85 years at baseline. In addition, the effects of potential predictors on sub-types of dementia (Alzheimer’s disease and vascular dementia) were also examined.

Analyses were performed using Stata 15.0 software (StataCorp, College Station, TX, USA). All reported *p* values were two-tailed, and the threshold for significance was *p* < 0.05.

## 3. Results

### 3.1. Demographic and Clinical Characteristics of the Study Cohort

We identified 341,910 people eligible for the medical care system at entry. Of these, 54,819 people with codes for dementia at baseline were excluded. We also excluded 60,353 people who died or moved out within 5 years of the observation period, for analytical purposes (Figure 1). Ultimately, data from 226,738 people were analyzed (baseline characteristics are shown in Table 1). Mean age was 80.0 years in men and 81.3 years in women. Males were more likely to have a history of cerebrovascular disease, diabetes mellitus, ischemic heart disease, and atrial fibrillation, compared to females. Females were more likely than males to have a history of hypertension and dyslipidemia, and to be prescribed antidepressants, antipsychotics, anxiolytics, and hypnotics.

### 3.2. Characteristics of Incident Dementia Cases

We identified 26,092 people who were diagnosed with dementia during the study period; 19,093 were classified as Alzheimer’s disease (ICD-10: F00, G30), 604 as vascular dementia (F01), and 5875 as unspecified dementia (F03) (Table 2). The overall prescription prevalence of anti-dementia drugs was 63%, and prevalence by age showed a decreasing trend with increasing age. Prescriptions of anti-dementia agents also differed by type of dementia, with those with Alzheimer’s disease having the highest rate of prescriptions (78%), and rates of anticholinesterase and memantine prescriptions of 67% and 19%, respectively (Table 2).

### 3.3. Logistic Regression Analysis

Table 3 shows the results of multivariable logistic regression analyses of predictors of a new diagnosis of dementia. Cerebrovascular disease (odds ratio (OR), 1.15; 95% confidence interval (CI), 1.11–1.18), depression/antidepressant use (OR, 1.38; 95% CI, 1.31–1.44), antipsychotic use (OR, 1.40; 95% CI, 1.31–1.49), hypnotic use (OR, 1.17; 95% CI, 1.11–1.24), and antithrombotic use (OR, 1.06; 95% CI, 1.02–1.10) were positive predictors of a new diagnosis of dementia, whereas antihypertensive drug use (OR, 0.90; 95% CI, 0.88–0.93) and dyslipidemia drug use (OR, 0.92; 95% CI, 0.89–0.95) were negative predictors of dementia in the multivariable-adjusted model. Multivariable adjustments for possible confounders only marginally altered the results of the age- and sex-adjusted model. Additional analysis revealed that the above-mentioned items were also predictors of a new diagnosis of Alzheimer’s disease (Appendix A), while cerebrovascular disease (OR, 2.20; 95% CI, 1.83–2.65), depression/antidepressant use (OR, 1.33; 95% CI, 1.02–1.74), and antipsychotic use (OR, 2.25; 95% CI, 1.64–3.07) were identified as positive predictors of a new diagnosis of vascular dementia (Appendix A).

The age-group analysis (75–79, 80–84, and ≥85 years) revealed that the predictive ability of cerebrovascular disease, diabetes mellitus, and use of psychiatric medications for a new diagnosis of dementia decreases with age (Table 4).

## 4. Discussion

In this study, the analyses of claims data from 226,738 community-dwelling elderly individuals in Niigata prefecture in Japan revealed a significantly higher risk of new dementia diagnoses among those with comorbidities of cerebrovascular disease, as well as those who are prescribed antidepressants, antipsychotics, or hypnotics, especially in the 75–79-year age group. According to the type of dementia, cerebrovascular disease and antipsychotic use were common significant predictors for Alzheimer’s disease and vascular dementia, with a stronger association observed with vascular dementia. Depression/antidepressant use was also a common significant predictor for Alzheimer’s disease and vascular dementia. These results suggest the usefulness of administrative claims data for identifying new dementia diagnoses.

The present study found neurological or psychiatric factors to be the most positively associated predictors across dementia sub-types and age groups, with significant associations observed for depression and cerebrovascular disease. Cerebrovascular disease has been shown to increase the risk of developing dementia [12]. In most cases, dementia has a mixed pathology of both vascular and neurodegenerative factors [13,14]. In addition, cerebrovascular disease can promote Alzheimer’s disease and vice versa, resulting in a reciprocal interaction amplifying the respective pathogenic effects [15,16]. Thus, controlling the established cerebrovascular risk factors can be beneficial as it reduces the risk of subsequent dementia [17]. We found that depression was a predictor for dementia, which is consistent with previous reports [6,18]. However, whether depressive symptoms are an early sign of dementia or an independent risk factor for dementia remains unclear, and the present findings of 4-year follow-up do not provide insight as to whether depressive symptoms preceded the diagnosis of dementia. A recent study has suggested that depression is potentially preventable and may be a modifiable risk factor of subsequent dementia [19]. Longer follow-up is necessary to clarify these aspects.

With regard to cardiovascular risk factors, current use of antihypertensive drugs and dyslipidemia drugs negatively predicted subsequent dementia diagnosis among people aged ≥75 years. This finding is consistent with previous reports [20]. There is sufficiently strong evidence that the management of cardiovascular risk factors is a promising approach to prevent dementia [3]. Previous longitudinal studies have shown that the timing of exposure to vascular predictors may be critical. Midlife—but not late-life—hypertension is reportedly associated with a higher risk of cognitive decline [21], and this risk may be reduced by treatment of hypertension [6,22]. We found similar results, which reinforced the hypothesis that the association between cardiovascular risk factors and dementia differed by the age of the target population.

Consistent with results from previous studies [6,23,24], we found that diabetes mellitus was a significant predictor for developing dementia. However, this association was only observed in individuals aged 75–79 years (Table 4), suggesting that predictors of dementia may vary by age. The present analyses of administrative claim data did not clarify whether sustained hyperglycemia, long-term use of anti-diabetic drugs, or repeated drug-induced hypoglycemia are primary causes leading to the development of dementia. Further elucidation of the causality and appropriate clinical measures will require longitudinal and detailed clinical data, including those on the status of medication and blood glucose levels.

A meta-analysis of observational studies using routine clinical data from Western countries showed that neuropsychiatric symptoms including depression, anxiety, antipsychotic use, and history of stroke were positively associated with all dementia [25]. On the other hand, among a few studies using only administrative claim data, one exploratory case-control study reported that schizophrenia, depression, and stroke increased subsequent dementia risk [26]. The present study using claims data identified dementia predictors that were consistent with those reported previously. Routinely collected clinical data, including healthcare claims, will help in estimating population burden of dementia for policy making, as well as for practical decision making.

The strength of this study is the use of comprehensive prefecture-level claims data of people over 75 years of age. These data cover the total population, including people who live in institutions such as assisted living or nursing homes, which enhanced our study generalizability. On the other hand, this study had several limitations. First, administrative claims contain diagnoses that are coded for the reimbursement of medical services. In other words, dementia might be underreported because only diagnoses related to an actual treatment are recorded; therefore, those with dementia who did not consult a doctor could not be identified. To address this issue, we adopted conservative criteria for the outcome definition, reducing the possibility of overestimation and increasing the validity of the outcome. Second, we did not completely differentiate between dementia subtypes, as is often the case with studies using claims data [11]. Vascular dementia might be under-reported due to the lack of a definite therapeutic agent, and might also be misclassified as unspecified dementia. Third, these data were only from fiscal years 2012 and 2016, while those from fiscal years 2013–2015 were not available due to administrative reasons. Therefore, individuals who were newly diagnosed with dementia between 2013–2015 and subsequently died or emigrated before 2016 could not be assessed. Fourth, there may be residual confounding factors. We adjusted for age, sex, cardiovascular risk factors, and neurological or psychiatric risk factors, but other predictors of dementia (e.g., educational level, ApoE4 allele carrier status) could not be integrated into the analysis because our dataset lacked detailed sociodemographic parameters and genetic information. Fifth, the lack of clinical data such as blood pressure, blood glucose, and blood lipid levels might have limited the validity of causal inference of hypertension, diabetes mellitus, dyslipidemia, and medication for these diseases for developing subsequent dementia.

## 5. Conclusions

The analyses of administrative claims data revealed that comorbidities of cerebrovascular disease and depression, antipsychotic use, and hypnotic use are predictors of new dementia diagnoses in the general population of people aged ≥75 years.

## Figures and Tables

**Figure 1 ijerph-18-00629-f001:**
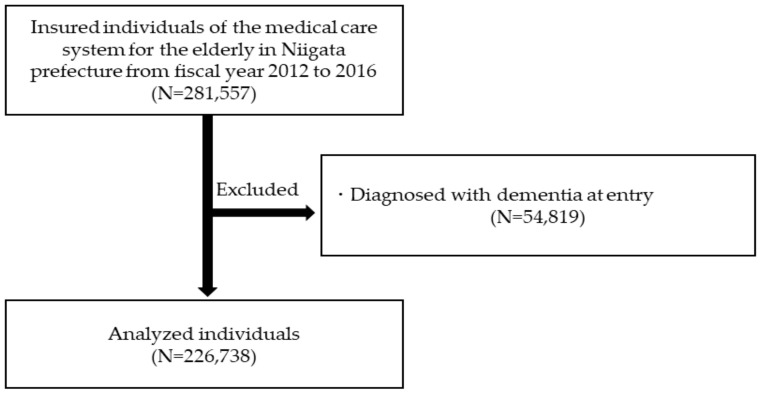
Flowchart of participant selection.

**Table 1 ijerph-18-00629-t001:** Baseline characteristics of the study cohort by sex.

Characteristics	Total		Male		Female	
	*n* = 226,738		*n* = 82,145		*n* = 144,593	
Baseline age, mean (SD)	80.81	(4.67)	80.02	(4.18)	81.26	(4.87)
Comorbidities, n (%)						
Cerebrovascular disease	61,979	(27.3%)	22,889	(27.9%)	39,090	(27.0%)
Diabetes mellitus	63,406	(28.0%)	26,202	(31.9%)	37,204	(25.7%)
Ischemic heart disease	37,955	(16.7%)	14,838	(18.1%)	23,117	(16.0%)
Atrial fibrillation	17,520	(7.7%)	8866	(10.8%)	8654	(6.0%)
Depression	16,845	(7.4%)	3828	(4.7%)	13,017	(9.0%)
Hypertension	154,132	(68.0%)	54,477	(66.3%)	99,655	(68.9%)
Dyslipidemia	98,723	(43.5%)	28,848	(35.1%)	69,875	(48.3%)
Prescriptions, n (%)						
Antidepressants	10,272	(4.5%)	2248	(2.7%)	8024	(5.5%)
Antipsychotics	7344	(3.2%)	2084	(2.5%)	5260	(3.6%)
Anxiolytics	53,707	(23.7%)	13,195	(16.1%)	40,512	(28.0%)
Hypnotics	18,704	(8.2%)	4428	(5.4%)	14,276	(9.9%)
Antihypertensive drugs	148,663	(65.6%)	52,575	(64.0%)	96,088	(66.5%)
Dyslipidemia drugs	68,662	(30.3%)	17,735	(21.6%)	50,927	(35.2%)

Abbreviation: SD, standard deviation.

**Table 2 ijerph-18-00629-t002:** Characteristics of incident dementia cases.

Type of Dementia	Alzheimer’s Disease	Vascular Dementia	Unspecified Dementia	All-Cause Dementia
Dementia incidence, n	19,093	604	5875	26,092
Age, n (%)				
75–79 y	6591 (34.5%)	201 (33.3%)	1603 (27.3%)	8627 (33.1%)
80–84 y	7447 (39.0%)	211 (34.9%)	1990 (33.9%)	9835 (37.7%)
≥85 y	5055 (26.5%)	192 (31.8%)	2282 (38.8%)	7630 (29.2%)
Prescriptions, n (%)				
Any anti-dementia drugs	14,912 (78.1%)	128 (21.2%)	1110 (18.9%)	16,330 (62.6%)
Anticholinesterases	12,851 (67.3%)	98 (16.2%)	886 (15.1%)	14,006 (53.7%)
Memantine	3670 (19.2%)	41 (6.8%)	318 (5.4%)	4050 (15.5%)

**Table 3 ijerph-18-00629-t003:** Multivariable logistic regression analyses of predictors for new diagnosis of dementia.

Predictors	Age- and Sex- Adjusted Model		Multivariable-Adjusted Model ^a^	
	OR (95%CI)	*p*-value	OR (95%CI)	*p*-value
Age (per 1-year increase)	1.06 (1.06–1.07) ^b^	<0.001	1.06 (1.06–1.07)	<0.001
Female	1.19 (1.16–1.22) ^c^	<0.001	1.18 (1.14–1.21)	<0.001
Cerebrovascular disease	1.19 (1.16–1.23)	<0.001	1.15 (1.11–1.18)	<0.001
Diabetes mellitus	1.02 (0.99–1.05)	0.152	1.03 (1.00–1.06)	0.075
Ischemic heart disease	1.00 (0.97–1.04)	0.857	0.99 (0.95–1.02)	0.430
Atrial fibrillation	0.98 (0.93–1.02)	0.321	0.97 (0.92–1.02)	0.233
Depression or use of antidepressants at baseline	1.56 (1.50–1.63)	<0.001	1.38 (1.31–1.44)	<0.001
Use of antipsychotics at baseline	1.71 (1.61–1.82)	<0.001	1.40 (1.31–1.49)	<0.001
Use of anxiolytics at baseline	1.15 (1.11–1.18)	<0.001	1.00 (0.96–1.03)	0.851
Use of hypnotics at baseline	1.33 (1.27–1.38)	<0.001	1.17 (1.11–1.24)	<0.001
Use of antihypertensive drugs at baseline	0.94 (0.91–0.97)	<0.001	0.90 (0.88–0.93)	<0.001
Use of dyslipidemia drugs at baseline	0.94 (0.91–0.96)	<0.001	0.92 (0.89–0.95)	<0.001
Use of antithrombotics at baseline	1.12 (1.09–1.15)	<0.001	1.06 (1.02–1.10)	0.001

Abbreviations: OR, odds ratio; CI, confidence interval. ^a^ Adjusted for age, sex, cardiovascular risk factors, and neurological or psychiatric risk factors. ^b^ The OR of age was adjusted for sex. ^c^ The OR of sex was adjusted for age.

**Table 4 ijerph-18-00629-t004:** Multivariable logistic regression analysis of predictors for new diagnosis of dementia by age group.

	75–79 y	80–84 y	≥85 y
Predictors	OR (95% CI)	OR (95% CI)	OR (95% CI)
Age (per 1-year increase)	1.14 (1.12–1.16)	1.08 (1.06–1.10)	0.99 (0.99–1.00)
Female	1.19 (1.14–1.25)	1.23 (1.17–1.29)	1.12 (1.06–1.19)
Cerebrovascular disease	1.25 (1.18–1.32)	1.13 (1.08–1.19)	1.06 (1.00–1.13)
Diabetes mellitus	1.12 (1.06–1.17)	0.98 (0.93–1.03)	0.97 (0.91–1.03)
Ischemic heart disease	0.99 (0.92–1.05)	1.01 (0.95–1.07)	0.96 (0.90–1.03)
Atrial fibrillation	1.06 (0.97–1.16)	0.95 (0.87–1.02)	0.87 (0.79–0.96)
Depression or use of antidepressants at baseline	1.47 (1.36–1.59)	1.38 (1.28–1.49)	1.21 (1.10–1.32)
Use of antipsychotics at baseline	1.48 (1.33–1.64)	1.40 (1.25–1.56)	1.31 (1.15–1.49)
Use of anxiolytics at baseline	1.03 (0.97–1.10)	0.95 (0.89–1.01)	0.97 (0.91–1.05)
Use of hypnotics at baseline	1.17 (1.07–1.28)	1.19 (1.10–1.30)	1.16 (1.04–1.29)
Use of antihypertensive drugs at baseline	0.89 (0.85–0.94)	0.89 (0.85–0.93)	0.90 (0.85–0.95)
Use of dyslipidemia drugs at baseline	0.87 (0.83–0.92)	0.92 (0.88–0.97)	0.96 (0.91–1.02)
Use of antithrombotics at baseline	1.09 (1.02–1.15)	1.05 (0.99–1.11)	1.01 (0.94–1.07)

Abbreviations: OR, odds ratio; CI, confidence interval.

## Data Availability

Restrictions apply to the availability of these data. Data was obtained from the Niigata wide area union of the medical care system for the elderly in Japan and are not publicly available. Data are however available from the authors upon reasonable request and with the permission of the Niigata wide area union.

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
