# Peer review of "Predictors of New Dementia Diagnoses in Elderly Individuals: A Retrospective Cohort Study Based on Prefecture-Wide Claims Data in Japan"

_ijerph, 2021, doi:10.3390/ijerph18020629_

Round 1

Reviewer 1 Report

This is retrospective cohort study examines possible predictors of new dementia diagnoses in a large group of elderly individuals.

The study represents a useful contribution, but it does have shortcomings:

  1. Lines 65-67 and 131-133: It would be helpful to indicate the criteria by which the ICD and ACT codes are determined. It would also be helpful to include the drugs covered by the various ACT codes.
  2. It would be helpful if the authors could indicate the duration of drug treatment, as much as is possible.
  3. Line 115: with the large number of comparisons conducted, the risk of generating false positives is high. Steps need to be taken to mitigate this risk. It seems that parametric statistics have been performed – have the authors assessed the assumptions undelaying this method?
  4. Line 128: Table 1 is slightly misleading as row 1 indicates an n number, but the actual n for all of the variables in column 1 has to be lower than the n in row 1. This needs to be corrected.
  5. Tables 3 and 4: There is evidence to indicate that a large number of people of dementia have both Alzheimer’s disease and vascular dementia. Therefore, it would be good to know whether cardiovascular disease is a predictor of Alzheimer’s disease.
  6. I suspicious about the effect of hypnotics. How many people were on hypnotics? What hypnotics were they given? How long for? The effect size is small, so is this false positive data.
  7. How did the authors explain antipsychotic treatment being a predictor of dementia?

Reviewer 2 Report

Overall, I felt this was a very sound study.  The large and nearly exhaustive prefecture-wide sample, of course, is impressive.  The methodology appears sound, and in particular I think it was wise to include in the dementia group only those diagnosed in two different billing quarters or in both inpatient and outpatient settings to exclude possible coding errors.  The Introduction makes a strong case for the value of the study.  The conclusions of the study follow logically, and the primary aim, to enhance the prediction of population burden of dementia, is appropriate in scope. 

I have only small suggestions, which I hope will add to the clarity of the presentation: 

Line 124:  "Females were significantly more likely than males to be prescribed antidepressants... and hypnotics."  If I am reading Table 1 correctly, it looks like they were also more likely to have a history of hypertension and dyslipidemia; my thought would be to mention this in the text. 

Lines 131 to 132:  It seemed surprising that of the 26,092 incident cases of dementia during the study, 73% (19,093) would be diagnosed as Alzheimer's disease and only 2% (604) as vascular.  Would this be expected for the prefecture?  I realize you have no control over the data, but I think a comment on this seeming disproportion in the Discussion would be helpful. 

Tables 3 and 4:  What is the direction of the odds ratio for sex, i.e., does the OR of 1.19 mean greater risk for women or for men?  Also, I take for granted that the OR of 1.06 per year of age refers to risk increasing with increasing age but I think this should be mentioned explicitly. 

Line 160:  I think it should be "predictive ability" rather than "predictability."  I believe "predictability" would mean how well cerebrovascular disease, e.g.,  can be predicted given a new diagnosis of dementia, as opposed to how well cerebrovascular disease, e.g., can predict dementia. 

I apologize for these rather trivial suggestions; they reflect the fact that I feel this manuscript is quite sound in all its major features. 
